# Cascaded Extended State Observer-Based Composite Sliding-Mode Controller for a PMSM Speed-Loop Anti-Interference Control Strategy

**DOI:** 10.3390/s25041133

**Published:** 2025-02-13

**Authors:** Yifan Xu, Bin Zhang, Yuxin Kang, He Wang

**Affiliations:** 1Changchun Institute of Optics, Fine Mechanics and Physics, Chinese Academy of Sciences, Changchun 130033, China; xuyifan22@mails.ucas.ac.cn (Y.X.); kangyuxin22@mails.ucas.ac.cn (Y.K.); wanghe22@mails.ucas.ac.cn (H.W.); 2University of Chinese Academy of Sciences, Beijing 100049, China

**Keywords:** permanent-magnet synchronous motor (PMSM), improved sliding-mode control (ISMC), new reaching law (NRL), continuous adaptive fast terminal sliding-mode surface (CAFTSMS), quasi-proportional resonant (QPR), cascaded extended state observer (CESO)

## Abstract

To enhance the speed-control performance of a permanent magnet synchronous motor (PMSM) drive system, an improved sliding-mode anti-interference control strategy is presented. Firstly, to tackle the speed fluctuation issue caused by cogging torque (a periodic disturbance) and time-varying disturbances at low set speeds in PMSM, an improved sliding-mode control (ISMC) is proposed. It consists of a continuous adaptive fast terminal sliding-mode surface (CAFTSMS) and a new reaching law (NRL). The CAFTSMS boosts the system’s immunity to interference, while the NRL, improved via an adaptive function, enhances the fast transient response and notably reduces speed fluctuations. Secondly, a quasi-proportional resonant (QPR) controller is introduced. It suppresses specific-order system harmonics, significantly reducing the harmonic amplitude and strengthening the system’s ability to handle periodic disturbances. Finally, a cascaded extended state observer (CESO) with a special cascade structure is proposed to solve the observation-delay problem in the traditional cascade structure. Experimental results show that the proposed sliding-mode anti-disturbance control strategy performs excellently in overcoming disturbances.

## 1. Introduction

Permanent-magnet synchronous motors (PMSMs) have gained increasing popularity across diverse fields such as electric vehicles and large-aperture telescopes [1,2,3,4] due to their high efficiency, torque density, and accuracy. PMSMs typically adopt a field-oriented control (FOC) strategy with a double closed-loop configuration, and their performance is significantly influenced by the dynamics of the speed control loop [5].

Both internal and external disturbances in the PMSM system severely impact its dynamic and steady-state performance. Moreover, the PMSM drive system is a multivariable, strongly coupled, and nonlinear system. Although traditional linear proportional integral (PI) controllers can meet some control requirements, they fall short of providing the high-precision speed control needed when the system is subjected to uncertain disturbances like internal parameter variations and external load torque fluctuations [6]. Consequently, extensive research has been carried out on various speed control strategies for PMSMs using modern control theory. These include fuzzy control [7], neural network control [8], sliding-mode control (SMC) [9,10,11,12], and active disturbance rejection control (ADRC) [13,14,15]. Among these, SMC is particularly favored in PMSM drive systems because it is independent of precise system models and highly resilient to external disturbances, significantly enhancing the system’s ability to withstand disruptions.

However, traditional SMC has issues such as sliding-mode chattering [16]. To further improve the speed and anti-interference performance of traditional SMC, numerous studies have focused on the sliding-mode surface and reaching law [17,18,19]. In [20], the smoothed sliding-mode control (SSMC) method was adopted for the first time. It mitigated chattering by replacing the sign function in sliding-mode control with a saturation function, but it did not take into account the issue of convergence speed. Ref. [21] adopted the fast terminal sliding-mode control (CFTSMC). This method had faster error convergence, higher tracking accuracy, and strong robustness against load disturbances, but it did not have a significant effect on suppressing speed chattering. In their research, Kom et al. [22] introduced a novel exponential reaching law (ERL) method for sliding-mode control. Different from traditional reaching law techniques, this ERL method did not use a sign function, resulting in a shorter convergence time. However, it only considered the issue of convergence speed and did not take chattering suppression into account.

This paper improves the sliding-mode surface and reaching law of traditional SMC. The enhanced sliding-mode surface improves the system’s dynamic response and interference resistance. The new reaching law (NRL), incorporating adaptive function ideas, significantly increases the system’s convergence speed and dynamic performance. Additionally, due to the use of the adaptive function, the improved sliding-mode control (ISMC) presented in this paper enables the system to have a faster response speed and less speed fluctuation. Many researchers have conducted extensive studies to enhance control strategies for suppressing periodic disturbances and specific frequency resonances. Reference [23] proposed an approach to improve active disturbance rejection control (ADRC) for suppressing periodic disturbances in the current loop by combining self-oscillating and quasi-resonant (QR) controllers. The work in [24] introduced an extended state observer (ESO) to reduce periodic torque ripples by integrating a traditional innovative ESO with QR controllers. In [25], a QR controller was proposed to work with a modified ADRC system to reduce the primary harmonic component of torque ripple, facilitating smoother speed control. This strategy effectively suppressed torque ripples because the QR controller can generate a high control gain at the resonant frequency. Furthermore, ref. [26] presented an adaptive observer based on quasi-proportional resonance (QPR) to achieve a reaction potential with adaptive harmonic cancellation. It eliminating the error between the observed and actual currents in the αβ reference system and enhancing observation performance. Finally, ref. [27] discussed combining a QPR controller with a super-twisted extended state observer (STESO) to reduce motor speed fluctuations.

This study incorporates the QPR concept to enhance the attenuation of harmonic components in specific frequency bands related to particular order disturbances, thus improving the system’s ability to suppress periodic disturbances. The implementation of the ESO enhances the resilience of control systems against time-varying disturbances. There are two main strategies to improve the ESO’s ability to detect and mitigate these perturbations. The first is to increase the observer’s bandwidth, which minimizes estimation errors and improves dynamic responsiveness. Researchers have proposed various bandwidth adjustment methods, such as bandwidth-optimized deep-reinforcement learning [28] and adaptive bandwidth techniques [29]. The second strategy focuses on modifying the ESO’s structure. For example, elevating the ESO order vertically to effectively track sinusoidal disturbances is called the generalized ESO (GESO) [30]. Alternatively, augmenting the ESO levels horizontally to enhance disturbance-estimation performance is known as the cascaded ESO (CESO) [31].

In this paper, we present a novel ESO cascade structure. Compared with the cascade structure in [32], this new design significantly reduces the pressure of selecting the observer bandwidth and addresses the issue of observation delay.

Our contributions can be summarized as follows:The ISMC, which is constructed based on a CAFTSMS and NRL, exhibits a strong anti-interference ability. It simultaneously enhances the response speed and effectively reduces the speed fluctuation value.We introduced QPR into the controller to suppress specific harmonic orders and reduce the harmonic amplitude.The proposed cascaded ESO has strong observational performance, is well aware of the observational error, and solves the problem of observation delay.The control strategy based on QPR-ISMC and CESO provides a control system with a strong anti-interference ability.

The subsequent chapters are organized as follows: Section 2 provides an overview of the mathematical model of the permanent magnet synchronous motor (PMSM); Section 3 details the design of the improved sliding-mode control (ISMC), along with a stability proof; Section 4 introduces a quasi proportional resonant (QPR) controller, and Section 5 introduces the cascaded extended state observer (CESO). Section 6 discusses the experimental results, and Section 7 concludes the paper.

## 2. PMSM Mathematical Model

One of the primary sources of periodic disturbances in the speed control loop of a permanent magnet synchronous motor (PMSM) is cogging torque. This phenomenon arises from the tangential component of the interaction force between the stator armature teeth and the permanent magnets within the PMSM. Specifically, cogging torque is generated by the interaction between the permanent magnets and the iron core when the motor windings are de-energized. Numerous studies conducted by researchers have indicated that cogging torque is contingent upon the rotor position of the PMSM and exhibits periodic fluctuations, with a period defined as the least common multiple $N_{c}$ of the number of stator slots and the number of rotor poles in the PMSM. The expression for the cogging torque is as follows:(1)$$T_{\mathrm{cog}}=\sum_{n=1}^{\infty }T_{n}\sin\left(N_{c}\theta_{e}\right)$$
where *n* is a positive integer, $T_{n}$ is the amplitude of the n -th cogging torque component, and $\theta_{e}$ is the rotor electrical angle. Cogging torque induces variations in both the torque and rotational speed of PMSM, significantly impairing the motor’s control performance, particularly at lower speeds [33].

By neglecting the effects of core saturation, eddy currents, and hysteresis losses on the PMSM, the voltage equation of a surface-mounted PMSM in a synchronous rotating (d−q) coordinate system can be expressed as follows:(2)$$\left\{\begin{array}{c} u_{d}=Ri_{d}+L\frac{di_{d}}{dt}-P_{n}\omega_{m}Li_{q}\\ u_{q}=Ri_{q}+L\frac{di_{q}}{dt}+P_{n}\omega_{m}\left(Li_{d}+\varphi_{f}\right) \end{array}\right.$$
where $u_{d}$, $u_{q}$, $i_{d}$, and $i_{q}$ represent the voltage and current components in the *d* and *q* axes, respectively, *R* represents the stator resistance, *L* is the stator inductance, $\omega_{m}$ represents the mechanical angular velocity of the rotor, $P_{n}$ represents the polar logarithm, and $\varphi_{f}$ represents the permanent magnet chain.

The kinetic model of a PMSM can be described as follows:(3)$${\dot{\omega }}_{m}=\frac{K_{t}}{J}i_{q}-\frac{B}{J}\omega_{m}-\frac{T_{L}}{J}$$
where *J* is the moment of inertia, $T_{L}$ is the load torque, *B* is the viscosity coefficient, and $K_{t}=\frac{3P_{n}\varphi_{f}}{2}$ is the torque constant.

Considering the effect of motor parameter variations on the motor performance, the dynamic model of the PMSM can be described as follows:(4)$$\left(J+\Delta \tau \right)\frac{d\omega_{m}}{dt}=\left(K_{t}+\Delta \kappa \right)i_{q}-\left(B+\Delta \rho \right)\omega_{m}-T_{L}$$
where Δτ, Δκ, and Δρ represent the parameter variations of the rotational inertia, damping coefficient, and torque constant, respectively.

Furthermore, the kinetic Equation (Equation 3) for a PMSM can be rewritten as follows:(5)$${\dot{\omega }}_{m}=\frac{K_{t}}{J}i_{q}^{ref}+D_{\omega }$$
where $i_{q}^{ref}$ denotes the *q*-axis reference current. $D_{\omega }$ is the aggregate disturbance, formulated as follows:(6)$$D_{\omega }=\left(\frac{K_{t}+\Delta \kappa }{J+\Delta \tau }\right)i_{q}-\frac{K_{t}}{J}i_{q}^{ref}-\left(\frac{B+\Delta \rho }{J+\Delta \tau }\right)\omega_{m}-\frac{T_{L}}{J+\Delta \tau }.$$

## 3. ISMC Proposal and Stability Demonstration

For sliding-mode control (SMC), the phase trajectory can be divided into two phases: arriving and the sliding.

### 3.1. Design of Traditional Sliding-Mode Control

First, the velocity error $x_{1}$ is defined as follows:(7)$$x_{1}=\omega_{m}^{ref}-\omega_{m}$$
where $\omega_{m}^{ref}$ is the given mechanical angular velocity of the rotor. Combined with Equation (Equation 5), $x_{1}$ and $x_{2}$ can be derived as follows:(8)$$\left\{\begin{array}{c} {\dot{x}}_{1}=-{\dot{\omega }}_{m}=-\frac{K_{t}}{J}i_{q}^{ref}-D_{\omega }\\ {\dot{x}}_{2}={\ddot{x}}_{1}=-\frac{K_{t}}{J}u. \end{array}\right.$$
where, $u=\dot{i_{q}^{ref}}$.

Subsequently, the sliding-mode surface *s* is designed as follows:(9)$$s=x_{1}+cx_{2}$$
where c>0. The traditional exponential reaching law is expressed as follows:(10)$$\frac{ds}{dt}=-\varepsilon sgn\left(s\right)-ks$$
where εsgn(s) represents the isochronous convergence term, and *ks* represents the exponential convergence term. The derivation of Equation (Equation 9) results in the following:(11)$$\dot{s}=c{\dot{x}}_{1}+{\dot{x}}_{2}=cx_{2}+{\dot{x}}_{2}=cx_{2}-\frac{K_{t}}{J}u=-\varepsilon sgn\left(s\right)-ks.$$

It can be further obtained that(12)$$u=\frac{J}{K_{t}}\left[cx_{2}+\varepsilon sgn\left(s\right)+ks\right]$$
and(13)$${\dot{x}}_{2}=-\frac{K_{t}}{J}u=-cx_{2}-\varepsilon sgn\left(s\right)-ks$$

According to Equations (Equation 8), (Equation 12) and (Equation 13), the reference current along the q-axis is as follows:(14)$$\begin{array}{cc} i_{q}^{ref} & =-\frac{J}{K_{t}}\left({\dot{x}}_{1}+D_{\omega }\right)\\ \; & =\frac{J}{K_{t}}\left\{\int_{0}^{t}\left[cx_{2}+\varepsilon sgn\left(s\right)+ks\right]dt-D_{\omega }\right\}. \end{array}$$

In our study of traditional sliding-mode control algorithms, we found that simply using a linear sliding-mode surface prevented the system variables from maintaining a sliding-mode, regardless of their initial conditions. The traditional isochronous reaching method solves the reachability problem by increasing the isochronous convergence term. However the discontinuity inherent in this convergence term leads to high-frequency oscillations in the sliding-mode surface. In addition, an increase in the gain of the exponential convergence term exacerbates the sliding-mode oscillations induced by the isochronous convergence term, even though the overall system convergence rate of increases. This analysis lays the foundation for developing a new sliding-mode control that incorporates improved sliding-mode surfaces and reaching laws.

### 3.2. Design of the Improved Sliding-Mode Control

To address the above issues, the continuous adaptive fast-terminal sliding-mode surface (CAFTSMS) was designed as follows:(15)$$s=x_{2}+\alpha_{1}{\left|x_{1}\right|}^{\sigma_{1}}sgn\left(x_{1}\right)+\alpha_{2}{\left|x_{2}\right|}^{\sigma_{2}}sgn\left(x_{2}\right)$$
where $x_{1}$, $x_{2}$ are consistent with the selection of Equations (Equation 7) and (Equation 8). $\alpha_{1},\alpha_{2}>0$. To further simplify the parameters, $\sigma_{1}=\frac{q_{x}}{p_{x}},\sigma_{2}=\frac{p_{x}}{q_{x}}$, $p_{x}<2q_{x}$ and $p_{x}>q_{x}$. $p_{x},q_{x}$ are odd.

To reduce the sliding-mode chattering, the exponential reaching law (ERL) proposed in this study uses the terminal attraction term instead of the conventional isochronous term. An adaptive function was incorporated to enhance the convergence speed of the system. The design of the NRL is as follows:(16)$$\dot{s}=-\alpha \frac{1+\lambda -e^{-a|s|}}{\lambda }sgn\left(s\right)-\beta {\left|s\right|}^{b}s$$
where α,β,λ ,and a>0; 0<b<1.

For the first term in NRL (Equation 16), the rate convergence rate is contingent on the characteristics of the sliding-mode function *s*. When $\left|s\right|$ tends to infinity, $\left(1+\lambda -e^{-a\left|s\right|}\right)/\lambda \approx \left(1+\lambda \right)/\lambda$. The enhanced reaching law exhibits a switching gain that surpasses the constant gain associated with the conventional isochronous term.

Consequently, the sliding-mode function *s* is expected to converge to zero at an accelerated rate owing to the increased switching gain; as $\left|s\right|$ converges to 0, $(1+\lambda -e^{-a\left|s\right|})/\lambda \approx 1$. The enhanced reaching law switching gain converges towards the constant gain characteristic of the conventional isochronous term.

Compared to the traditional isochronous convergence gain term, the parameter ε can be assigned a reduced sliding-mode chattering. Additionally, a smaller value of λ can accelerate the convergence and enhance the dynamic response capabilities of the controller; however, excessively small values of λ may result in overshooting.

The adaptive function can be clearly observed in the context of the second term of the proposed NRL (Equation 16). It can be guarantees a greater gain when the system deviates from the sliding-mode surface, thereby enhancing the system convergence rate, as indicated by expression ${\left|s\right|}^{b}$. Simultaneously, a smaller gain is provided when the system approaches the sliding-mode surface, decreasing the system convergence speed. The first and second terms in the NRL, with the addition of the adaptive function in this study are compared with the conventional SMC as shown in Figure 1. The horizontal axis of the coordinates represents the sliding-mode function *s* and the vertical axis represents the speed of convergence to the sliding-mode surface.

According to Equations (Equation 15) and (Equation 16), the total output *u* of the controller can be expressed as follows:(17)$$u=\frac{J}{Kt}\left(u_{1}+u_{2}\right)$$

Among them(18)$$u_{1}=x_{2}+\alpha_{1}|x_{1}{|}^{\sigma_{1}}sgn\left(x_{1}\right)+\alpha_{2}{\left|x_{2}\right|}^{\sigma_{2}}sgn\left(x_{2}\right)$$(19)$$u_{2}=\int_{0}^{t}(\alpha \frac{1+\lambda -e^{-a|s|}}{\lambda }sgn\left(s\right)+{\beta |s|}^{b}s)dt.$$

Eventually we can obtain the following:(20)$$i_{q}^{ref}=u-\frac{J}{Kt}\hat{D}\left(t\right)=\frac{J}{Kt}\left(u_{1}+u_{2}-\hat{D}\left(t\right)\right).$$

The control block diagram of the ISMC is shown in Figure 2.

### 3.3. ISMC Stability Demonstration

*Proof* 1: The Lyapunov function was chosen as: $V=\frac{1}{2}s^{2}$. Taking the derivative of *V* and combining it with Equation (Equation 16) yields the following:(21)$$\dot{V}=s\dot{s}=s(-\alpha \frac{1+\lambda -e^{-a|s|}}{\lambda }sgn\left(s\right)-{\beta |s|}^{b}s).$$

Because of(22)$$-\alpha \frac{1+\lambda -e^{-a|s|}}{\lambda }sgn\left(s\right)s\leq 0$$
and(23)$$-{\beta |s|}^{b}s^{2}\leq 0$$

Then, we obtain(24)$$\dot{V}\leq 0$$

According to Lyapunov’s convergence theorem, since(25)V≥0

And, if and only if s=0, then V=0. The system can be guaranteed to converge in finite time.

## 4. Introduction of the QPR Controller

Given the adverse effects linked to speed variations, a widely utilized QPR controller was employed to reduce the harmonic components in this scenario. In contrast to conventional PR controllers, the QPR controller examined in this research is less sensitive to fluctuations in the resonant frequency. As a result, the QPR controller retains its effectiveness in mitigating harmonic components, even in instances where the resonant frequency is inaccurately defined. The transfer function of the QPR controller is expressed in the following manner:(26)$$G_{\mathrm{QPR}}\left(s\right)=\frac{y(s)}{x(s)}=k_{p}+\frac{2k_{r}\omega_{c}s}{s^{2}+2\omega_{c}s+\omega_{r}^{2}}.$$

In this context, $\omega_{r}$ represents the resonant frequency, while $\omega_{c}$ denotes the bandwidth of the QPR controller. Additionally, $k_{p}$ and $k_{r}$ signify the proportional and resonant coefficients, respectively. The bode diagrams of the QPR controller with different parameters are shown in Figure 3. Each of the four sets of simulations changes only a single parameter in the QPR controller: (a) $k_{r}$ = 10, $\omega_{c}$ = 4%$\omega_{r}$, $\omega_{r}$ = 200, $k_{p}$ = (1 to 3). (b) $k_{p}$ = 2, $k_{r}$ = 10, $\omega_{c}$ = 4%$\omega_{r}$, $\omega_{r}$ = (100 to 300). (c) $k_{p}$ = 2, $k_{r}$ = 10, $\omega_{r}$ = 200, $\omega_{c}$ = (2%$\omega_{r}$ to 6%$\omega_{r}$). (d) $k_{p}$ = 2, $\omega_{c}$ = 4%$\omega_{r}$, $\omega_{r}$=200, $k_{r}$ = (5 to 15).

The QPR controller demonstrates a significant control gain at frequency $\omega_{r}$, which effectively reduces the harmonic component associated with that frequency. Drawing from the frequency domain characteristics depicted in Figure 3, a framework for parameter selection is established for the QPR controller. It is crucial to select a sufficiently large parameter $k_{r}$ to alleviate the torque ripple at frequency $\omega_{r}$. In contrast, adjusting parameter $\omega_{c}$ affects the operational range of the QPR controller near the resonant frequency, thereby mitigating the risk of system instability that may result from an excessively high value of $k_{r}$.

## 5. Introduction of CESO

### 5.1. Design of the Traditional Extended State Observer (ESO)

The state-space representation of the speed control loop for A PMSM is expressed as follows:(27)$$\left\{\begin{array}{c} \frac{dx}{dt}=Ax+Bu\\ y=Cx \end{array}\right.$$
where $x={\left[\begin{array}{cc} \omega_{m} & D_{\omega } \end{array}\right]}^{T}$ is the state vector of the system, $y=\omega_{m}$ is the output of the system, $u=i_{q}^{ref}$ is is the input of the system. And $A=\left|\begin{array}{cc} 0 & 1\\ 0 & 0 \end{array}\right|,\;B=\left[\begin{array}{c} \frac{K_{t}}{J}\\ 0 \end{array}\right],\;C=\left[\begin{array}{cc} 1 & 0 \end{array}\right].$

The design of the traditional extended state observer (ESO) is as follows:(28)$$\left\{\begin{array}{c} \frac{d\hat{x}}{dt}=Ax+Bu+H\varphi \\ \hat{y}=C\hat{x} \end{array}\right.$$
where $H=[h_{1}$$h_{2}{]}^{T}$ is the gain matrix of the observer, $\hat{x}$ is the observed value of the state vector*x*, and $\hat{y}$ is the observed value of the system output *y*. From Equations (Equation 27) and (Equation 28), the derivative of the observation error $\tilde{x}=\hat{x}-x$ can be obtained as follows:(29)$$\frac{d\tilde{x}}{dt}=\left(A-HC\right)\tilde{x}.$$

To ensure the observer stability, it is necessary to select an appropriate observer–system matrix to ensure that the real parts of the eigenvalues are negative; that is(30)Re{λi(A−HC)}<0i=1<1,2.

Assume that $l_{1}=l_{2}=-\omega_{0}\left(\omega_{0}>0\right)$, then $H=[2\omega_{0}$${\omega_{0}}^{2}{]}^{T}$ can be solved. Moreover from Equation (Equation 35), we can obtain the following:(31)$${\dot{\hat{D}}}_{\omega }={\omega_{0}}^{2}\left(\omega_{m}-{\hat{\omega }}_{m}\right)$$
where $\omega_{0}$ is the bandwidth of the observer.

### 5.2. Design of the CESO

To enhance the disturbance attenuation, a cascade structure-based CESO has been proposed. The CESO consists of two consecutive layers, referred to as ESO (1) and ESO (2). Concerning ESO (1):(32)$$\left\{\begin{array}{c} {\dot{z}}_{11}=-l_{11}\left(\omega_{m}-z_{11}\right)+\frac{K_{t}}{J}i_{q}^{ref}+z_{12}\\ {\dot{z}}_{12}=-l_{12}\left(\omega_{m}-z_{11}\right) \end{array}\right..$$

In addition, ESO (2) can be designed according to(33)$$\left\{\begin{array}{c} {\dot{z}}_{21}=-l_{21}\left(\omega_{m}-z_{21}\right)+z_{22}+z_{12}+\frac{K_{t}}{J}i_{q}^{ref}\\ {\dot{z}}_{22}=-l_{22}\left(\omega_{m}-z_{21}\right) \end{array}\right..$$

According to the bandwidth method, to achieve observer convergence, the following must be satisfied:(34)$$\left\{\begin{array}{c} l_{11}=l_{21}=2\omega_{0}\\ {l_{12}=l_{22}=\omega_{0}}^{2} \end{array}\right.$$

The final CESO output ${\hat{\omega }}_{m}$, ${\hat{D}}_{\omega }$ is(35)$$\left\{\begin{array}{c} {\hat{\omega }}_{m}=z_{21}\\ {\hat{D}}_{\omega }=z_{12}+z_{22} \end{array}\right..$$

From Equation (Equation 35), it is apparent that $z_{22}$ functions as an additional estimate derived from $z_{12}$, specifically representing a further evaluation of the perturbation based on $z_{12}$. It is crucial to emphasize that the second cascaded extend state observer (CESO) analyzed in this research is explicitly designed to monitor the output $x_{1}$ of the system. The innovative special cascade structure significantly reduces the delay associated with the subsequent cascade, thereby improving the estimation efficiency. The control block diagram for the CESO is depicted in Figure 4 and Figure 5 present the block diagram representing the speed regulation system of the PMSM.

### 5.3. Stability Analysis of the CESO

The estimation error of CESO1 can be derived as follows:(36)$$\left\{\begin{array}{c} {\dot{e}}_{11}={\dot{z}}_{11}-{\dot{x}}_{1}=e_{12}-\beta_{11}e_{11}\\ {\dot{e}}_{12}={\dot{z}}_{12}-{\dot{x}}_{2}=-\beta_{12}e_{11}-\varphi . \end{array}\right.$$

Let $\eta_{i}=e_{1i}/\omega_{o}^{i-1},i=1,2$, and then, Equation (Equation 36) can be rewritten as follows:(37)$$\dot{\eta }=\omega_{o}H\eta +\varphi K/\omega_{o}$$
where,$$\eta =\left[\begin{array}{c} \eta_{1}\\ \eta_{2} \end{array}\right],\;H=\left[\begin{array}{cc} -2 & 1\\ -1 & 0 \end{array}\right],\;K=\left[\begin{array}{c} 0\\ -1 \end{array}\right].$$

Both eigenvalues of *H* are −1, so *H* is hurwitz stable. Then, there exists a unique positive definite matrix *P* such that(38)$$H^{T}P+PH=-I$$
where $P=\left[\begin{array}{cc} \left(1/2\right) & -(1/2)\\ -(1/2) & \left(3/2\right) \end{array}\right].$ Select the Lyapunov function as $V\left(\eta \right)=\eta^{T}P\eta$, and then,(39)$$\dot{V}\left(\eta \right)={\dot{\eta }}^{T}P\eta +\eta^{T}P\dot{\eta }=-\omega_{o}{∥\eta ∥}^{2}+2\omega_{o}^{-1}\eta^{T}PK\varphi .$$

Since φ is globally Lipschitz in terms of *x*, that is, there exists a constant ξ such that $\varphi \leq \xi ∥x-z_{1i}∥$ for all $x,z_{1i}$, one has(40)$$2\omega_{o}^{-1}\eta^{T}PK\varphi \leq 2\zeta \omega_{o}^{-1}\eta^{T}PK∥x-z_{1i}∥.$$

When $\omega_{o}\geq 1$, one has $\omega_{o}^{-1}∥x-z_{1i}∥=\omega_{o}^{-1}∥e_{1i}∥\leq ∥\eta ∥.$ Thus, we have(41)$$2\omega_{o}^{-1}\eta^{T}PK\varphi \leq \delta {∥\eta ∥}^{2}$$
where $\delta ={∥PK\xi ∥}^{2}+1.$ Substituting (30) into (28), we can obtain(42)$$\dot{V}\left(\eta \right)\leq -\left(\omega_{o}-\delta \right){∥\eta ∥}^{2}$$
that is, $\dot{V}\left(\eta \right)<0$ if $\omega_{o}>\delta .$ Similarly, let $e_{2}i=z_{2i}-x_{i}$ as the estimation error of the CESO2, and the same derivation as Equation (Equation 42) can be obtained. Therefore, we obtain(43)$$\lim_{t\to \infty }e_{{\,}_{1i}}=0,\;\lim_{t\to \infty }e_{{\,}_{2i}}=0,\;i=1,2.$$

## 6. Experimental Studies

### 6.1. Experimental Condition

Correlation studies were conducted under constant speed, variable speed, and sud-den load conditions using a 2.2-kW surface permanent-magnet synchronous motor (SPMSM) experimental setup. This setup comprised a drive motor, load motor, DC power source, motor driver, encoder, PC, load controller, and main controller, which utilized a TI TMS320C28346 (Texas Instruments, Texas, USA) floating-point processor. A physical representation of the experimental platform is illustrated in Figure 6, in which a dual motor-to-drag loading mode was employed. The nominal specifications of the SPMSM are detailed in Table 1.

In the controller experiment for the CAFTSMS introduced in this study, the parameters were set as follows: $\alpha_{1}=0.5$, $\alpha_{2}=1.7$, $p_{x}=5$, and $q_{x}=3$. For the NRL proposed in this study, the parameters were defined as α=3, λ=0.4, a=1.7, β=1.6, and b=0.26. For the QPR introduced in this study, the parameters were defined as follows: $k_{r}=10$, $k_{p}=1$, $\omega_{r1}$ = 11, $\omega_{c1}$ = 4%$\omega_{r1}$, and $\omega_{r2}=0.5\omega_{r1}$. For the PI, the parameters were defined as $k_{p}=0.1$, and $k_{i}=0.5$, and the parameters of the SMC are c=0.35, k=5, ε=35. The current loops used PI controllers with the parameters $k_{p}=2$, and $k_{i}=100$.

### 6.2. Performance of ISMC

In the first set of experiments to verify the superiority of the proposed ISMC, three control experiments using conventional PI and SMC were completed under low speed-setting conditions.

In the first experiment, the speed is set to 100 rpm. From Figure 7, it can be observed that the response times of the three control algorithms, PI, SMC, and ISMC, are 4.75 s, 2.86 s, and 1.83 s, respectively. The ISMC presented in this study demonstrates a notable advantage in terms of the response speed. The peak speed fluctuations in the steady-state phase of the three control algorithms are 4 r, 3.48 r, and 2.44 r. The proposed ISMC has less speed fluctuation in the steady-state phase, owing to the removal of the isochronous term applied in the conventional SMC. The fast Fourier transform (FFT) analysis of the speed also shows that the harmonic orders of the speed are mainly in the first and second orders and the proposed ISMC has the lowest amplitude. Meanwhile, the steady-state rotational speed of the ISMC proposed in this paper has a smaller variance.

In the second experiment, the speed was set to 50 rpm. As shown in Figure 8, the peak speed demonstrates in the steady-state phases of the PI, SMC, and ISMC were 6.17 r, 7.32 r, and 2.74 r, respectively. The ISMC proposed in this study also had less speed demonstrated in the steady-state phase. An FFT analysis of the experimental speed data with a set speed of 50 r also yields similar results to the 100 r experiment: the proposed ISMC exhibited the lowest amplitude at the first and second-order harmonics.

In the third experiment, the rotation speed was set to 75 rpm. From Figure 9, it can be observed that the peak speed demonstrate in the steady-state phases of the PI, SMC, and ISMC were 4.32 r, 2.76 r, and 1.88 r, respectively. The ISMC proposed in this study also had less speed demonstrate in the steady-state phase. The ISMC proposed in this paper also has less speed fluctuation value in the steady state phase. Figure 10 shows a comparison of the fluctuation peaks of the three algorithms at different speeds.

### 6.3. Harmonic-Suppression Capability of QPR-ISMC

In this set of experiments, a QPR controller was added to the control algorithm. As can be seen from Figure 11, the amplitudes of the ISMC at the first and second-order harmonic amplitudes are 0.20 and 0.29, respectively, when the QPR controller is not added. After applying the QPR controller to the second-order harmonic, it is evident that the amplitude at the first-order harmonic amplitude was reduced to 0.12. After suppressing both the first and second-order harmonics, it can be seen that the first-order harmonic amplitude decreases to 0.13 and the second-order harmonic amplitude decreases to 0.22. These are 35.00% and 24.14% decreases in the first- and second-order harmonic amplitudes, respectively, compared with the control algorithm without applying the QPR controller.

### 6.4. CESO Performance

In this study, two sets of experiments were conducted to verify the ability of the pro-posed improved sliding-mode control algorithm to overcome the time-varying disturbances. The first set of experimental control algorithms comprised the traditional SMC, ISMC without CESO, and ISMC with CESO. The experimental load torque was set to 2 N·m and 4 N·m, respectively.

Through Figure 12, we can clearly see that when the load torque is 2 N·m, the maximum rotational-speed fluctuations of the three control algorithms are 20.72 r, 8.30 r, and 6.35 r, respectively, with regulation times of 3.37 s, 1.25 s, and 1.13 s, respectively. When the load torque is 4 N·m, the maximum rotational-speed fluctuation values for the three control algorithms are 42.75 r, 9.55 r, and 9.35 r with regulation times of 4.36 s, 1.30 s, and 0.70 s, respectively. It can be concluded that the proposed ISMC has a strong anti-disturbance capability, and introducing CESO further enhances the system’s ability to overcome time-varying disturbances.

To verify the superiority of the proposed CESO over the traditional ESO, in the second set of experiments, applying the same load torque and selecting the same observer bandwidth $\omega_{o}$, the fluctuation situation of the observed values was observed. It can be seen from Figure 13 that when the same load was applied, for different magnitudes of the load set values, the peak values of the observation fluctuations of the CESO were, respectively, reduced by 31.51% and 20.34%, compared with those of the ESO. When the load was unloaded, they decreased by 40.79% and 28.85%, respectively. By comparison, we can conclude that the proposed CESO has a stronger observation ability than the traditional ESO.

The third set of experiments was conducted to verify the superior performance of the proposed CESO to observe the rotational speed. Through Figure 14, we can clearly see that the proposed CESO has a smaller observation error, has a strong ability to observe the rotational speed, and the peak of the observer’s observation-error fluctuation shrinks when a larger observer bandwidth $\omega_{0}$ is selected. However, it is worth noting that the peak of the observer’s observation-error fluctuation is reduced when a larger observer is selected. However, when a larger observer bandwidth $\omega_{0}$ is selected, the observation-error fluctuation value in the steady-state phase becomes larger, which suggests that we should reasonably select the controller bandwidth $\omega_{o}$.

## 7. Conclusions

In this study, we proposed an ISMC composed of the NRL and CAFTSMS. The experimental results showed that the control algorithm proposed in this paper had a faster response speed with smaller sliding-mode jitter and stronger anti-interference capability. Moreover, the introduced QPR controller reduced the harmonic amplitude of the system at specific orders. Finally, the special cascade structure of the CESO further enhanced the system’s resistance to time-varying disturbances. In summary, the experimental results showed that the improved sliding-mode anti-interference control strategy proposed in this paper was highly superior. Future work may focus on the direction of expanding or adjusting the proposed method by integrating machine learning techniques to achieve the adaptive optimization of control parameters. Deep learning algorithms will be utilized to analyze a large amount of operation data, automatically adjusting the parameters of key components such as QPR and CESO to adapt to different operating conditions and system changes, and further enhancing the performance and adaptability of the control strategy.

## Figures and Tables

**Figure 1 sensors-25-01133-f001:**
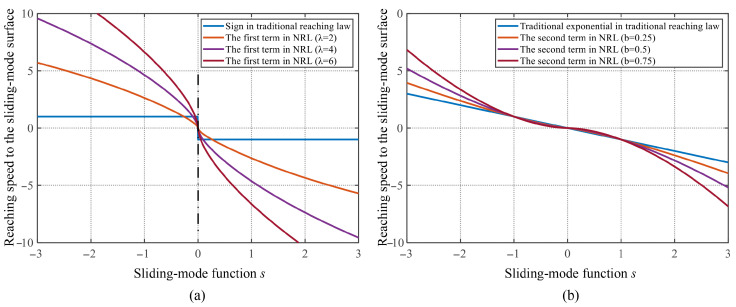
Comparison of the terminal attraction term with traditional sign function.

**Figure 2 sensors-25-01133-f002:**
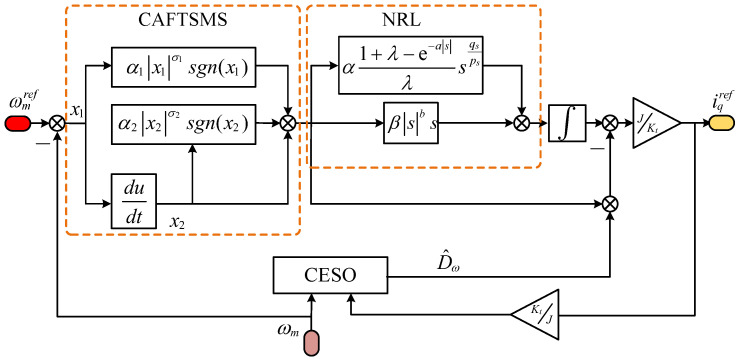
ISMC control-block diagram.

**Figure 3 sensors-25-01133-f003:**
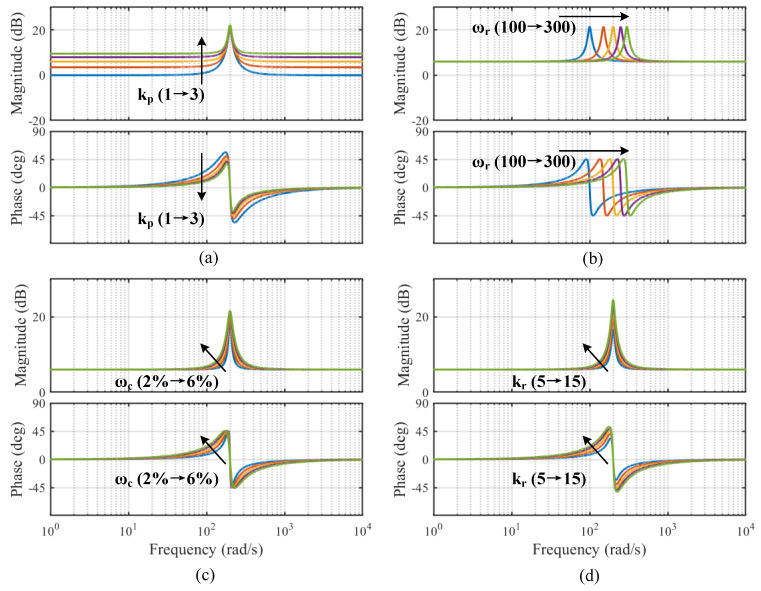
Bode diagrams of the QPR controller with one variable parameter. (**a**) $k_{p}$. (**b**) $\omega_{r}$. (**c**) $\omega_{c}$. (**d**) $k_{r}$.

**Figure 4 sensors-25-01133-f004:**
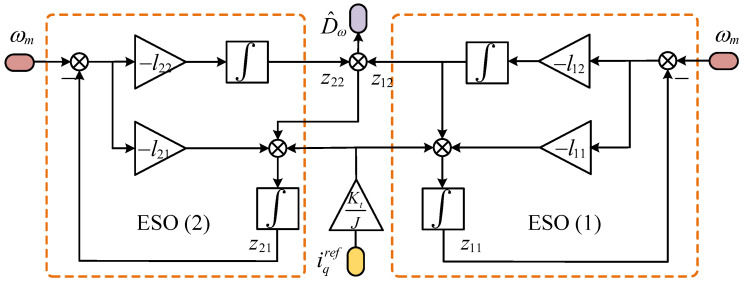
Control block diagram for the CESO.

**Figure 5 sensors-25-01133-f005:**
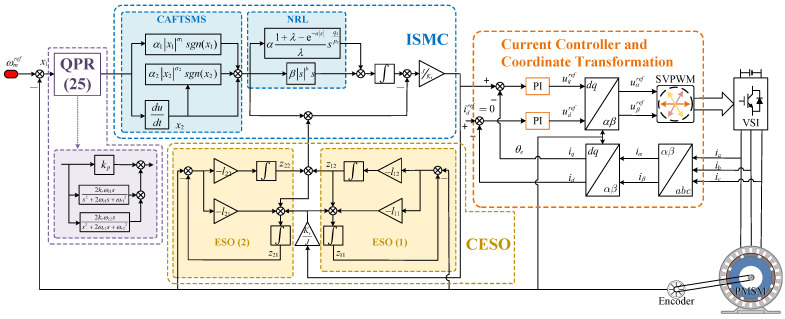
PMSM speed-regulation system.

**Figure 6 sensors-25-01133-f006:**
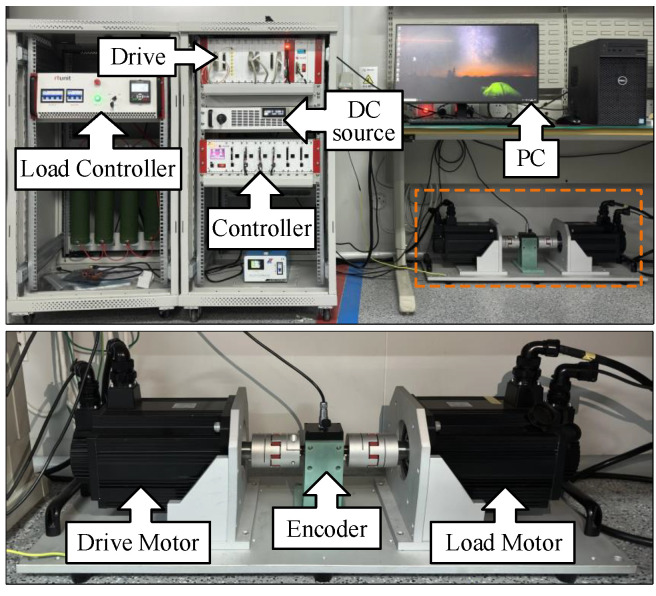
Experimental platform for permanent magnet synchronous motors.

**Figure 7 sensors-25-01133-f007:**
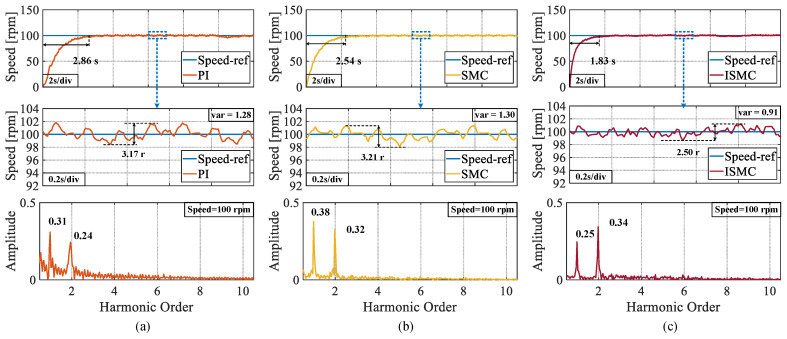
Step response of three algorithms at a set speed of 100 rpm, steady state speed fluctuation values and FFT analysis of speeds. (**a**) PI. (**b**) SMC. (**c**) ISMC.

**Figure 8 sensors-25-01133-f008:**
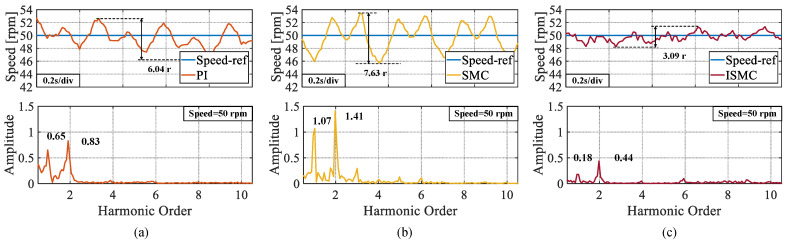
Steady state speed fluctuation values and FFT analysis of speeds at a set speed of 50 rpm. (**a**) PI. (**b**) SMC. (**c**) ISMC.

**Figure 9 sensors-25-01133-f009:**
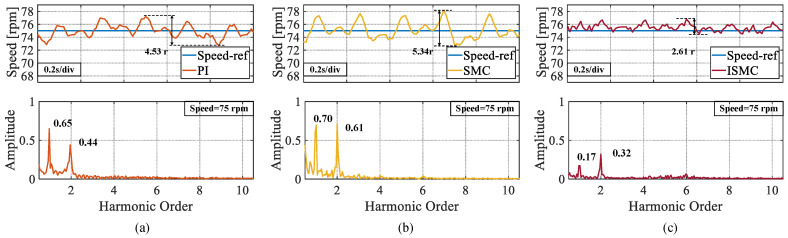
Steady state speed fluctuation values and FFT analysis of speeds at a set speed of 75 rpm. (**a**) PI. (**b**) SMC. (**c**) ISMC.

**Figure 10 sensors-25-01133-f010:**
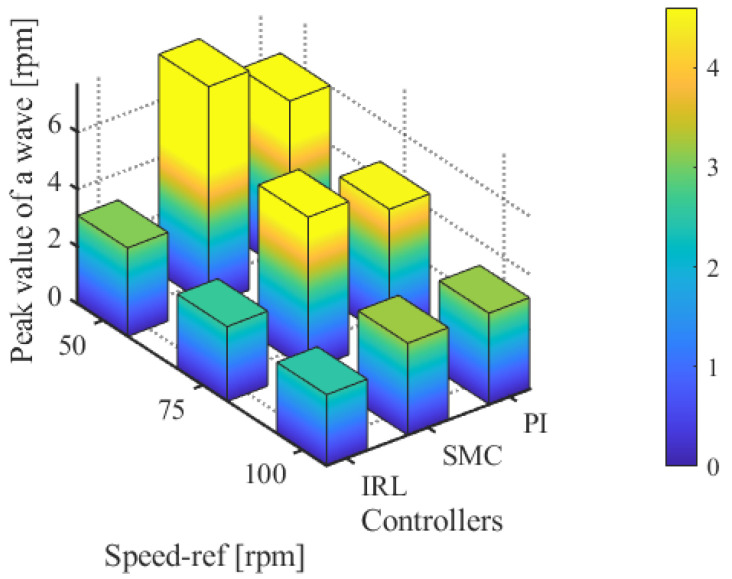
Comparison of maximum speed fluctuation values of three algorithms.

**Figure 11 sensors-25-01133-f011:**
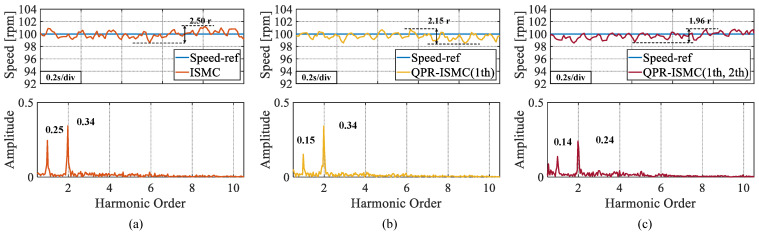
The experimental results of introducing QPR on the basis of ISMC. (**a**) No QPR introduced. (**b**) 1 th. (**c**) 1 th and 2 th.

**Figure 12 sensors-25-01133-f012:**
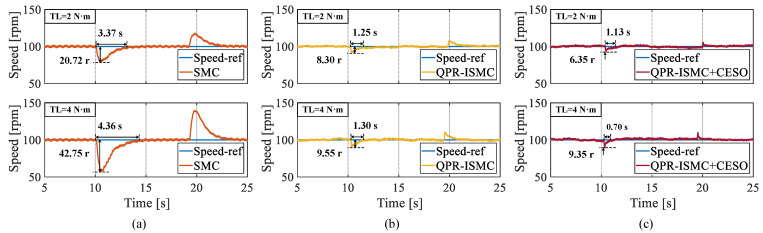
The experimental results of applying and unloading the same magnitude of load to three algorithms. (**a**) SMC. (**b**) QPR-ISMC. (**c**) QPR-ISMC+CESO.

**Figure 13 sensors-25-01133-f013:**
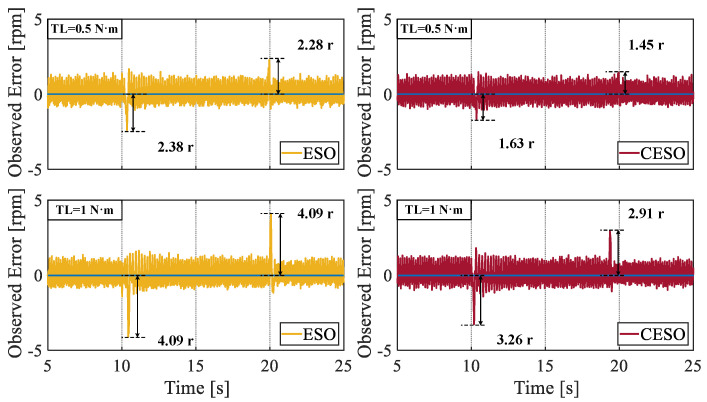
Comparison of the observation error between ESO and CESO.

**Figure 14 sensors-25-01133-f014:**
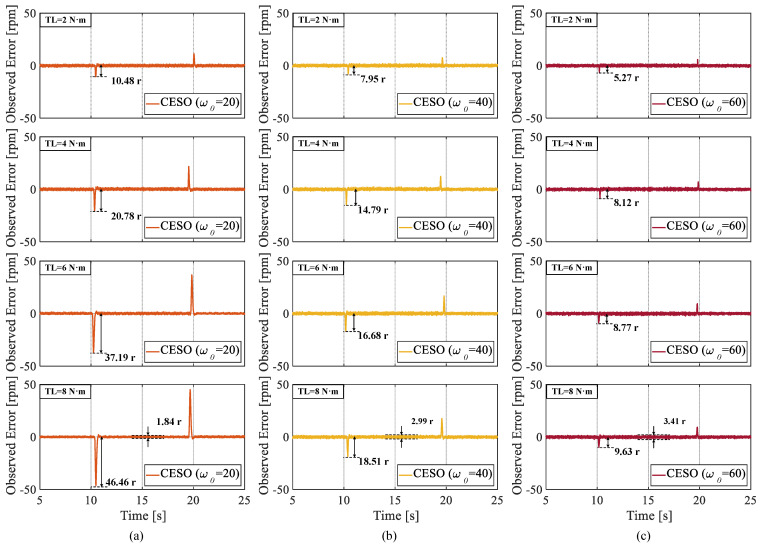
Observation error of the observer when different observer bandwidths $\omega_{0}$ are selected. (**a**) $\omega_{0}$ = 20. (**b**) $\omega_{0}$ = 40. (**c**) $\omega_{0}$ = 60.

**Table 1 sensors-25-01133-t001:** Values of the parameters in the PMSM.

Parameter	Value and Unit
Number of pole pairs	4
Stator inductance	6.5 mH
Stator resistance	0.12 Ω
Moment of inertia	0.028 $\mathrm{kg}\cdot m^{2}$
Damping factor	0.0048
Permanent-magnet flux linkage	0.18542 Wb

## Data Availability

Data is contained within the article.

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
