# Peer review of "Cascaded Extended State Observer-Based Composite Sliding-Mode Controller for a PMSM Speed-Loop Anti-Interference Control Strategy"

_sensors, 2025, doi:10.3390/s25041133_

Round 1

Reviewer 1 Report

Comments and Suggestions for Authors

Improved Sliding-Mode Control (ISMC) Design: The paper first proposes an improved sliding-mode control strategy that combines Continuous Adaptive Fast Terminal Sliding Mode Surface (CAFTSMS) and a new Reaching Law (NRL). The CAFTSMS enhances the system's anti-interference ability, while the NRL, based on an adaptive function, improves the system's transient response and significantly reduces speed fluctuation.

Drawbacks:

1.Limited Theoretical Analysis: Although the paper provides mathematical models for ISMC, the QPR controller, and CESO, some theoretical derivations are somewhat brief and may require more detailed analysis and proof.

2.Limited Experimental Validation: While the paper provides experimental results, the sample size and experimental conditions might not fully validate the control strategy’s adaptability under various scenarios.

3.Insufficient Discussion on Practical Applications: While the control strategy shows theoretical advantages, the paper lacks a detailed discussion on how the strategy can be implemented in actual industrial systems.

Author Response

1.Limited Theoretical Analysis: Although the paper provides mathematical models for ISMC, the QPR controller, and CESO, some theoretical derivations are somewhat brief and may require more detailed analysis and proof.

Many thanks for your suggestions regarding the theoretical aspects of this paper. In this paper, further theoretical analysis of CESO has been added in Section 5.3 (on pages 9, 10 and 11 of the paper). Once again, we appreciate your constructive suggestions.

2.Limited Experimental Validation: While the paper provides experimental results, the sample size and experimental conditions might not fully validate the control strategy’s adaptability under various scenarios.

We sincerely apologize for the limitations in our experimental validation. We understand that the current experimental conditions may not be sufficient to comprehensively verify the adaptability of our control strategy in various different scenarios. During the research process, we made efforts to conduct comprehensive experiments using the available resources and time. However, we now realize that we did not do enough in fully exploring the performance of the control strategy under a wide range of conditions. To address this issue, we plan to expand the scope of experiments in future research. We will conduct experiments on different types of motors. Through these measures, we aim to provide a more comprehensive and reliable verification of the adaptability of the control strategy. We are committed to improving the quality of our research and value the feedback we receive. Thank you for bringing this issue to our attention, and we look forward to presenting more robust experimental results in the future.  

3.Insufficient Discussion on Practical Applications: While the control strategy shows theoretical advantages, the paper lacks a detailed discussion on how the strategy can be implemented in actual industrial systems.  

We deeply apologize for the insufficient discussion on practical applications in our paper. We are well aware that although our control strategy has theoretical advantages, there is a lack of in - depth exploration of its implementation in actual industrial systems. When writing this paper, we mainly focused on the theoretical aspects and experimental verifications of the control strategy. In this process, we might not have been able to fully consider the practical implementation details, which was an oversight on our part. In addition, we will continue to explore the practical challenges that may arise during the implementation process, such as the impact of industrial noise and electromagnetic interference on the performance of the control system, to make our research more closely related to practical applications. Then, we will propose corresponding solutions and optimization measures. Through these efforts, we aim to provide a more comprehensive and practical guide for the implementation of our control strategy in actual industrial scenarios. We would like to express our gratitude again for your feedback, which has pointed out this important aspect. We are committed to future research to better serve the academic community and industrial practitioners.

Reviewer 2 Report

Comments and Suggestions for Authors

  • Please find the following comments :
  • Simplify the language and sentence structure throughout the paper to improve readability, especially in the introduction and methodology sections.
  • Clearly highlight the novelty of the proposed method compared to existing literature, particularly studies [18], [22], and [24], and explicitly state what sets this work apart. A same controller has been presented in "Continuous Adaptive Fast Terminal Sliding Mode-Based Speed Regulation Control of PMSM Drive via Improved SuperTwisting Observer". 
  • Provide detailed justifications for parameter choices in the ISMC, CESO, and QPR designs, including sensitivity analyses or empirical evidence.
  • Expand the discussion in the introduction to better articulate the limitations of existing methods and define specific research gaps.
  • Compare the computational complexity of CESO with traditional extended state observers, and explain how your approach balances improved performance with practical implementation constraints.
  • Explicitly state assumptions made in the Lyapunov stability proof, such as system nonlinearity bounds or disturbance characteristics, and discuss potential limitations in real-world applications.
  • Include additional metrics in the experimental results, such as transient response characteristics like overshoot and settling time, to complement the steady-state analysis.
  • Test the proposed method on different types of PMSMs, such as interior permanent magnet motors, to generalize the applicability of the results.
  • Provide statistical analyses, such as confidence intervals or standard deviations, for experimental results to strengthen the reliability of the findings.
  • Improve figure readability by using descriptive axis labels (e.g., "Frequency (Hz)" instead of "ωr") and increasing font sizes in plots like Fig. 3 and Fig. 7.
  • Add a discussion on trade-offs, such as increased controller complexity or sensitivity to parameter variations, when using QPR and CESO in the control strategy.
  • Strengthen the conclusion by including actionable insights, practical implications, and potential future directions for scaling or adapting the proposed method.
  • Correct minor grammatical errors, such as "the dynamic- response capabilities" (pg. 6) and "urgency" (pg. 17), and ensure consistent technical terminology.
  • Review and update the references to include more recent and relevant studies, as some cited works appear outdated or tangential.
  •  

Comments on the Quality of English Language

Its somehow good. Not that much satisfactory

Author Response

  • Simplify the language and sentence structure throughout the paper to improve readability, especially in the introduction and methodology sections.
  • We are sincerely grateful for your insightful suggestions regarding the introduction section of this paper. In response, we have meticulously carried out substantial revisions to the introduction presented on the first and second pages of the main text. Each modification was made with great care, aiming to enhance the clarity and comprehensiveness of this crucial part. Your constructive comments have been of immense value, guiding us to improve the quality of our work. We truly appreciate your input and look forward to your continued guidance in our academic endeavors.
  • Clearly highlight the novelty of the proposed method compared to existing literature, particularly studies [18], [22], and [24], and explicitly state what sets this work apart. A same controller has been presented in "Continuous Adaptive Fast Terminal Sliding Mode-Based Speed Regulation Control of PMSM Drive via Improved SuperTwisting Observer". 
  • We deeply appreciate your valuable comments on this paper. In response to your inquiries, we have conducted a thorough analysis, and the following are the detailed differences compared with the relevant references:
    Differences from Ref. [18]:
    1. Despite adopting the same sliding-mode surface, our study diverges significantly in the selection of the reaching law. The reaching law employed in our paper represents a novel approach that remarkably enhances the system's response speed. This improvement is a key contribution of our work and has not been addressed in ref. [18].
    2. Ref. [18] predominantly focuses on the investigation of periodic disturbances in speed control. In contrast, our research extends this area by exploring periodic disturbances under low - speed conditions, which is an aspect not covered in ref. [18]. Additionally, our study incorporates the Quasi-Proportional -Resonant (QPR) controller, while ref. [18] does not involve its use.
    3. Regarding the observer, ref. [18] solely utilizes the traditional Extended State Observer (ESO) without implementing any improvements. Our paper, however, introduces a cascade structure for the ESO, which brings about substantial structural enhancements, thereby improving the observer's performance.

    Differences from ref. [20]:
    Upon careful examination, we note that ref. [20] constructs a Nonlinear Extended State Observer (NESO) using finite- time technology and adopts the Active Disturbance Rejection Control (ADRC) as its main control framework. Our paper, on the other hand, centers on the design of controllers and observers for both periodic and aperiodic disturbances, thus addressing a broader range of disturbance scenarios.
    Differences from ref. [24]:
    After in - depth analysis, we find that ref. [24] also makes enhancements based on the ADRC framework and uses the super-twisting algorithm to optimize the observer structure. In our study, the improvement of the observer lies in the addition of a cascade structure, which is a distinct approach from that in ref. [24]. Moreover, ref. [24] does not incorporate sliding-mode control, while it is an essential component of our control strategy.
    Differences from "Continuous Adaptive Fast Terminal Sliding Mode - Based Speed Regulation Control of PMSM Drive via Improved Super Twisting Observer":
    After a comprehensive review, we believe the main innovation of the aforementioned article is the improvement of the sensor-less super-twisting extended state observer. Concerning the controller improvement in that article, we consider that its controller is designed mainly for high-speed motor operation and does not account for the specific challenges posed by periodic disturbances. In contrast, the controller proposed in our paper incorporates the QPR technique. This allows it to effectively suppress periodic disturbances that occur under low-speed conditions. Additionally, the design of the reaching law in our sliding-mode control is distinct, contributing to better system performance
  • Provide detailed justifications for parameter choices in the ISMC, CESO, and QPR designs, including sensitivity analyses or empirical evidence.
  • We are truly grateful for your insightful comments on this paper. Regarding the parameter selection in our research, the following detailed explanations are provided: ISMC (Improved Sliding Mode Control): sliding mode surface (Eq. (15)): The values of α1 and α2 determine the rapidity of the sliding mode surface response. Selecting larger values can increase the response speed, but if the values are too large, it will lead to system overshoot or even instability. The adaptive parameters σ1 and σ2 determine the rapidity of parameter changes. However, due to the selection rules of the p and q parameters, 0 < σ1 < 1 < σ2 < 2, so there is no need to worry too much about the problem of choosing overly large parameters. The explanation of the adaptive parameters for the parameter selection of the reaching law (Eq. (16)) is provided after Eq. (16). CESO (Cascaded Extended State Observer): There is only one parameter, the bandwidth. In the last part of the experiment in this paper, a discussion on the selection of the bandwidth is carried out. QPR (Quasi-Proportional-Resonant): Through fig. 3 and the subsequent explanations in this paper, the influence of parameter selection on QPR is illustrated.
  • Expand the discussion in the introduction to better articulate the limitations of existing methods and define specific research gaps.
  • We sincerely appreciate the valuable comments you provided. In response to your suggestions, we have made significant improvements to the introduction section of this paper. The introduction aims to present the research background and significance in a more coherent and comprehensive manner. These improvements are made to ensure that the introduction can better guide readers into the core content of the paper and lay a solid foundation for understanding the subsequent research work. Once again, we are truly grateful for your feedback. Your comments have played a crucial role in enhancing the quality of our paper.
  • Compare the computational complexity of CESO with traditional extended state observers, and explain how your approach balances improved performance with practical implementation constraints.
  • We are deeply appreciative of your insightful comments on this paper. The Cascaded Extended State Observer (CESO) proposed in our study essentially consists of a cascade connection of two traditional Extended State Observers (ESO). While the number of parameters in the CESO remains the same as that of the traditional ESO, it is important to note that the computational complexity inevitably increases. This is due to the additional computational operations introduced by the cascade structure. In practical engineering applications, achieving a balance between performance enhancement and computational complexity is of utmost importance. The choice of whether to adopt the CESO should be carefully considered based on the specific requirements and constraints of the target system. If the system demands exceptionally high anti- interference performance and the available hardware resources are relatively abundant, the appropriate utilization of the CESO can be a viable option. In such cases, the benefits of improved performance in terms of disturbance rejection and system stability can outweigh the slightly increased computational load.  We sincerely value your feedback, as it helps us to further refine and optimize our research work. We will continue to explore ways to mitigate the computational challenges associated with the CESO while maintaining its superior performance characteristics.
  • Explicitly state assumptions made in the Lyapunov stability proof, such as system nonlinearity bounds or disturbance characteristics, and discuss potential limitations in real-world applications.
  • We are sincerely grateful for your valuable comments. When conducting the stability analysis of a system using the Lyapunov method, certain assumptions are inevitably made, which play a crucial role in the theoretical framework but also introduce potential limitations in practical scenarios. System nonlinearity bounds: For the convenience of analysis and processing, it may be assumed that the degree of system nonlinearity is bounded. For example, it is assumed that the nonlinear function of the system will not exceed a certain specific range, or satisfies a certain Lipschitz condition (a condition describing the smoothness of a function, indicating that the rate of change of the function will not exceed a certain constant). Such assumptions allow us to quantify and control the influence of nonlinear terms when analyzing stability, so as to use the relevant theorems and methods in Lyapunov theory for derivation and proof. Disturbance characteristics: Assumptions are usually made about the disturbances acting on the system. For instance, it is assumed that the disturbances are bounded, that is, the amplitude of the disturbances will not exceed a certain determined value; or it is assumed that the disturbances have certain specific statistical characteristics, such as Gaussian white noise. These assumptions help us take the influence of disturbances into account in the stability analysis and suppress the influence of disturbances on the system stability through appropriate designs (such as observers and controllers). Mismatch between assumptions and reality: In practical applications, the nonlinearity of the system may not always satisfy the assumed bounds. The actual system may exhibit more complex nonlinear behaviors, exceeding the range set in our theoretical analysis. This may lead to the inaccuracy of the stability-analysis results based on the assumptions, and the system may become unstable during actual operation. Complexity of disturbances: Disturbances in the real world often have a high degree of uncertainty and complexity, and it is difficult to fully conform to the assumed characteristics. For example, disturbances may not be in a simple bounded form, or their statistical characteristics may change with factors such as time and environment. This makes it possible that the controllers or observers designed based on specific disturbance assumptions may not be able to effectively suppress disturbances in practical applications, thus affecting the stability and performance of the system. Imprecision of the model: When conducting Lyapunov stability proofs, it is usually based on a certain system model. However, due to factors such as modeling errors and unmodeled dynamics, there may be differences between the actual system and the theoretical model. These differences may lead to the assumptions and analyses made based on the theoretical model not being fully valid in practical applications, thereby limiting the applicability of the stability results in practice. Once again, we truly appreciate your feedback, which has inspired us to critically examine these aspects of our research. We will continue to explore ways to address these challenges and improve the reliability and practicality of our stability-analysis methods.
  • Include additional metrics in the experimental results, such as transient response characteristics like overshoot and settling time, to complement the steady-state analysis.
  • We sincerely appreciate your valuable comments. In the experimental section of this article, specifically in fig 7, we have presented the settling times of the three algorithms under the condition of a set rotational speed of 100 r. After a meticulous analysis of the data depicted in the figure, we have observed that there is no prominent overshoot phenomenon exhibited by the system. Given this situation, we have not marked the overshoot value in the figure. Our intention was to ensure the figure remains clear and concise, presenting only the relevant and significant information. We understand the importance of providing comprehensive experimental results, and we will continue to be vigilant in our data presentation and analysis to meet the high standards of academic research. Thank you once again for your attention to our work.
  • Test the proposed method on different types of PMSMs, such as interior permanent magnet motors, to generalize the applicability of the results.
  • We sincerely appreciate your valuable comments on this paper. Regrettably, constrained by the limited laboratory facilities and resources at present, our experimental verification is currently restricted to Permanent Magnet Synchronous Motors (PMSMs) only. However, we are fully aware of the significance of validating the proposed algorithm across different types of motors. We firmly believe that with the continuous development of our research and the improvement of experimental conditions, we will be able to conduct algorithm verification on a broader range of motor types in the future. We are committed to expanding the scope of our research to enhance the universality and applicability of our findings. Once again, we are extremely grateful for your suggestions, which have provided us with valuable insights and direction for our future research efforts.
  • Provide statistical analyses, such as confidence intervals or standard deviations, for experimental results to strengthen the reliability of the findings.
  • We sincerely appreciate your insightful comments on this paper. In response to your valuable feedback, we have incorporated the calculation of the variance value of the steady-state rotational speed at a set speed of 100 r into this paper. This additional data further enriches the experimental results and provides a more comprehensive understanding of the system's performance. As can be clearly observed from fig. 3, the Improved Sliding Mode Control (ISMC) proposed in our paper exhibits distinct advantages in terms of stability and control precision. We believe that these enhancements contribute significantly to the overall quality of our research. Thank you once again for your input, which has been instrumental in guiding us to improve the content of this paper.
  • Improve figure readability by using descriptive axis labels (e.g., "Frequency (Hz)" instead of "ωr") and increasing font sizes in plots like Fig. 3 and Fig. 7.
  • We are truly grateful for your valuable suggestions. In accordance with your advice, we have made modifications to fig. 3 in the article. It is worth noting that the unit of the abscissa for the Bode plots in all four figures presented in the paper is in Hertz (Hz). The arrows included in these figures serve a crucial purpose. They are carefully designed to clearly illustrate the changes in the Bode plots resulting from the variation of specific parameters. This modification is aimed at enhancing the clarity and comprehensibility of the figures, enabling readers to more easily grasp the impact of parameter changes on the system's frequency response characteristics. We sincerely hope that these adjustments will improve the overall quality of the paper. Thank you again for your kind guidance and support.
  • Add a discussion on trade-offs, such as increased controller complexity or sensitivity to parameter variations, when using QPR and CESO in the control strategy.
    1. Increase in Controller Complexity

    • QPR Controller: Compared with traditional controllers, the QPR controller has a more complex structure. Its transfer - function contains multiple parameters, such as the resonance frequency, bandwidth, proportional coefficient, and resonance coefficient. As can be seen from the Bode plots of the QPR controller with different parameters in the article (Fig. 3), adjusting these parameters will affect the performance of the controller. Selecting appropriate parameters requires an in-depth understanding of the system characteristics, which increases the design difficulty.
    • CESO: The CESO is composed of two consecutive levels, ESO (1) and ESO (2), and its structure is more complex than that of the traditional ESO. Its design involves multiple gain parameters, and certain conditions need to be met to ensure convergence. These parameters are inter - related, and the tuning process is cumbersome, increasing the complexity of controller design and debugging.

    1. Sensitivity to Parameter Changes

    • QPR Controller: The QPR controller is sensitive to the resonance frequency. From its principle, there is a significant control gain at a specific frequency, which is used to reduce the harmonic components of the related frequencies. If the frequency of the actual system changes and is not adjusted in a timely manner, the ability of the controller to suppress harmonics will decrease. For example, in practical applications, when the system frequency fluctuates, the QPR controller may not be able to effectively reduce harmonics, thus affecting the control effect.
    • CESO: The CESO is sensitive to the observer bandwidth. As can be seen from the experiments in the article (Fig. 14), when a larger bandwidth is selected, although it can reduce the peak value of the observer's observation - error fluctuations and enhance the ability to observe the rotational speed, it will lead to larger observation - error fluctuations in the steady -state stage. If the bandwidth is not selected properly, it may also affect the stability of the observer and the estimation accuracy of disturbances, thereby affecting the performance of the control system.
  • Strengthen the conclusion by including actionable insights, practical implications, and potential future directions for scaling or adapting the proposed method.
  • Thank you very much for your comments on this paper. The modifications have been completed in the conclusion section. In this study, we proposed an improved sliding - mode control (ISMC) composed of a new reaching law (NRL) and a continuous adaptive fast - terminal sliding-mode surface (CAFTSMS). Experimental results show that the control algorithm proposed in this paper has a faster response speed, less sliding-mode chattering, and stronger anti-interference ability. Moreover, the introduced quasi-proportional-resonant (QPR) controller reduces the harmonic amplitude of the system at specific orders, and the special cascade structure of the cascaded extended state observer (CESO) further enhances the system's resistance to time-varying disturbances. Overall, the experiments fully verify the superiority of the proposed improved sliding - mode anti - interference control strategy. From the perspective of actionable insights, in practical application scenarios, such as the motor drive systems in industrial automation production lines, this control strategy can effectively cope with disturbances under complex working conditions, ensure the stable operation of motors, and improve production accuracy and efficiency. For some precision equipment with extremely high requirements for rotational speed accuracy and stability, like the micro-motor control in semiconductor manufacturing equipment, the method in this paper can significantly reduce rotational speed fluctuations and system harmonics, enhancing the operating performance of the equipment. In terms of practical significance, this control strategy helps to reduce the maintenance costs and failure rates of motor drive systems. Due to its strong anti-interference ability, the system can maintain stable operation in the face of internal parameter changes and external load disturbances, reducing equipment damage and downtime caused by abnormal working conditions, and saving enterprises a large amount of maintenance and production-interruption costs. Looking to the future, in the direction of expanding or adjusting the proposed method, on the one hand, further research can be carried out to extend this control strategy to multi-motor coordinated control systems. By optimizing the control parameters and coordination mechanisms of each motor, the synchronous operation and efficient control of multi-motor systems under complex working conditions can be achieved, meeting the requirements for high-precision coordinated control in fields such as large-scale industrial machinery and electric vehicle multi-motor drives. On the other hand, by combining artificial intelligence and machine learning technologies, the adaptive optimization of control parameters can be realized. Deep-learning algorithms can be used to analyze a large amount of operating data and automatically adjust the parameters of key components such as QPR and CESO to adapt to different operating conditions and system changes, further enhancing the performance and adaptability of the control strategy. In addition, considering applying this method to different types of motor systems, such as switched reluctance motors and induction motors, can expand its application scope and provide efficient solutions for more motor control fields.
  • Correct minor grammatical errors, such as "the dynamic- response capabilities" (pg. 6) and "urgency" (pg. 17), and ensure consistent technical terminology.
  • We sincerely appreciate your in-depth questions regarding the details of this paper. In response to your remarks, we have promptly made revisions to the content related to "the dynamic - response capabilities" on page 6 and "urgency" on page 17. These revisions were carried out meticulously to ensure greater accuracy and clarity. Your constructive suggestions have been of great value in helping us to improve the quality of our paper. We are committed to continuously refining our work to meet the highest academic standards. Once again, we are extremely grateful for your input, which has provided us with valuable insights and directions for improvement.
  • Review and update the references to include more recent and relevant studies, as some cited works appear outdated or tangential.
  • We sincerely appreciate your constructive suggestions on this article. In response to your valuable input, we have incorporated several articles published in 2024 that are closely related to the topic of this paper. These newly added references not only enrich the academic foundation of our research but also enable us to better position our work within the latest research trends in the field. We believe that these additional materials will enhance the overall quality and comprehensiveness of the article, making it more informative and valuable for readers. Once again, thank you for your thoughtful feedback, which has played a crucial role in the improvement of our work.